# Distribution of Volatile Compounds in Different Fruit Structures in Four Tomato Cultivars

**DOI:** 10.3390/molecules24142594

**Published:** 2019-07-17

**Authors:** Jian Li, Taiju Di, Jinhe Bai

**Affiliations:** 1Beijing Engineering and Technology Research Center of Food Additives, Beijing Technology and Business University, 11 Fucheng Road, Beijing 100048, China; 2USDA-ARS, U.S. Horticultural Research Laboratory, 2001 South Rock Road, Ft. Pierce, FL 34945, USA

**Keywords:** tomato, volatile, aroma, different tissues

## Abstract

Distribution of volatile compounds in different fruit structures were analyzed in four tomato cultivars by headspace-solid-phase microextraction (SPME)-gas chromatography-mass spectrometry (GC-MS). A total of 36 volatile compounds were identified in fruit samples, which were primarily aldehydes, hydrocarbons, alcohols, ketones, furans, esters, nitrogen compounds, and sulfur and nitrogen-containing heterocyclic compounds. The volatile compositions in pericarp (PE), septa and columella (SC), locular gel and seeds (LS), and stem end (SE) tissues showed different profiles. The PE tissue showed the highest total volatile concentration due to a high abundance of aldehydes, especially cis-3-hexenal and benzaldehyde. Meanwhile, it showed higher aromatic proportion and herbaceous series intensity than other tissues. Floral and fruity series showed higher intensity in SC and LS tissues. The concentration of alcohols in the LS was higher than that in other tissues in association with the higher abundances of 2-methyl propanol, 3-methyl butanol, and 2-methyl butanol. However, the numbers and concentrations of volatile compounds, especially cis-3-hexenal, benzaldehyde, and geranyl acetone were lower in SE than in the other tissues, indicating less tomato aromas in SE. SE tissues were also lacking in floral and fruity characteristic compounds, such as geranyl acetone, 1-nitro-pentane, and 1-nitro-2-phenylethane. “FL 47” contained more volatile compounds than the other three, and the contents of aldehydes, ketones and oxygen-containing heterocyclic compounds in the “Tygress” fruit were higher than the other cultivars.

## 1. Introduction

Tomato is one of the world’s most consumed vegetables, and is known for its unique taste and richness of nutrients, such as lycopene, β-carotene, and lutein which can protect against various cardiovascular diseases and many forms of cancer [1].

In recent years, there has been an increasing demand for good quality fruit and vegetables, and the tomato is not an exception. More and more consumers are willing to pay extra for good flavor [1]. Fresh tomatoes have a characteristic flavor due to the presence of a complex mixture of sugars, acids, and volatile compounds [2,3]. Volatile compounds are the main determinant of the special flavor in tomatoes [4]. Scientists have identified more than four hundred volatile compounds in the tomato [5], but less than 10% of them are present in significant concentrations and odor thresholds for determining the tomato flavor [6,7]. In the past few years, many researchers studied the volatile compounds of the tomato and identified many biosynthetic pathways of essential aromatic volatiles [8]. The volatile compounds are biosynthesized from lipids, amino acids, lignins, and carotenoids [9].

Previously, many studies have been performed on the volatile components in whole fruit or pericarp tissues [10,11,12,13,14,15], and the contents of the volatile components in the peel and internal tissues were compared. Wang et al. [16] compared the composition of the volatiles between the pericarp and locular gel, and concluded that the locular gel contained lower concentrations of cis-3-hexenal, hexanal, heptanal, octanal, nonanal, cymene, terpinolene, undecane, dodecane, 6-methyl-5hepten-2-one, 2-methyl butyl acetate, 1-nitro-pentane, and 1-nitro-2-phenylethane than the pericarp. However, the locular gel contained higher concentrations of 2-methyl propanal, butanal, 2-methyl butanal, 2-methyl-2-butenal, 2-methyl propanol, 3-methyl butanol, 2-methyl butanol, and 2-butanone than the pericarp [16]. Wang et al. [17] also compared the differences in the volatile components between the peel and internal tissues, and showed that the inner tissues of the tomato fruit contained higher concentrations of 3-methyl butanal, 2-methyl butanal, 3-methyl butanol, and 2-methyl butanol than that of the pericarp, which is probably due to the higher levels of alcohols in the inner tissues [17,18].

Previous studies were focused on determining the volatile compositions only in the pericarp and locular gel. Actually, the anatomy of a tomato fruit includes pericarp (PE), septa and columella (SC), locular gel and seeds (LS), and the stem end (SE) (Figure 1), and the volatile compositions in the different inner tissues were not reported. The inner tissues also have a lot of volatile compounds, and they make great contributions to the overall aroma quality. In this study, the ripe tomato fruits of “FL47”, “Tygress”, “Tasti-Lee”, and “Cherokee Purple” cultivars were used. The volatiles were extracted from PE, SC, LS, and SE, and the organic volatile compounds were identified by Gas Chromatography-Mass Spectrometry (GC-MS). The aim of the paper was to analyze the distribution of volatile compounds in different fruit structures in four tomato cultivars and evaluate the aroma contributions and aroma profile of different tissues based on their odor activity values (OAVs), which would provide substantial information regarding the volatile components in different cultivars and inner tissues.

## 2. Results and Discussion

### 2.1. The Proportion of Each Tissue of the Four Tomato Varieties

Table 1 showed the weight of each tissue of the four cultivars of tomato used in this study. Higher proportions of the pericarp (PE) tissue samples were used among the four varieties, and the sampling proportions of “Cherokee Purple” were higher from septa and columella (SC) and stem end (SE), but lower from PE and locular gel and seeds (LS) tissues compared to the other three varieties—as a result, it had more flesh and was of better quality. The “Cherokee Purple” cultivar is a traditional variety that passed down through several generations of a family which possessed evolutionary resistance against pests and diseases, and has been adapted to specific growth conditions and climates [19]. Over the last 50 years, much of the focus was on yield, and important aspects of fruit quality were largely neglected [1], which resulted in a decline in quality. In the past 40 years, many Cherokee Purple varieties were lost or replaced by some commercially attractive hybrid tomatoes. Cross-breeding of crops has generated many high-yield varieties with low quality. Largely, the genetic and biochemical complexities of these traits have diminished their respective characteristics [20].

### 2.2. Volatile Compounds in Tomato Fruits

Table 2 exhibits the volatile compounds and their odor description, odor thresholds in water, and RI values. The solid-phase micro-extraction-gas chromatography-mass spectrometry (SPME-GC/MS) study identified 36 volatile compounds, including 13 aldehydes, 6 hydrocarbons, 5 alcohols, 4 ketones, 2 oxygen-containing heterocyclic compounds, 3 esters, 1 nitrogen compound, and 2 sulfur- and nitrogen-containing heterocyclic compounds. According to Table 3, aldehydes accounted for the highest percentage of the total volatile concentration, followed by alcohols and ketones. Among the individual compounds, cis-3-hexenal was the most abundant component, which agreed with the reports of Wang et al. [16], following trans-2-hexenal, hexanal, acetone, 2-methylbutanal, and 3-methylbutanal (Table 3).

Thirteen volatile compounds, which were suggested to be important tomato aroma contributors in previous studies, were identified in our study [4,21,22,23], including 3-methyl butanal, 2-methyl butanal, cis-3-hexenal, hexanal, trans-2-hexenal, 3-methyl butanol, 2-methyl butanol, 1-penten-3-one, 6-methyl-5-hepten-2-one, 2-isobutyl thiazole, 1-nitro-2-phenylethane, geranyl acetone, and methyl salicylate. The biosynthetic origins of the 13 volatiles were reported previously. Based on their biosynthetic origins, these compounds can be divided into four groups: fatty acid derivatives (cis-3-hexenal, hexanal, trans-2-hexenal, 1-penten-3-one), carotenoid derivatives (6-methyl-5-hepten-2-one, geranyl acetone), amino acid derivatives (3-methyl butanal, 2-methyl butanal, 3-methyl butanol, 2-methyl butanol, and 2-isobutyl thiazole), and phenylalanine derivatives (1-nitro-2-phenylethane and methyl salicylate) [1].

Odor threshold values in water, adapted from Wang et al. [17].

### 2.3. Differences in the Volatile Profiles among the Cultivars

The profiles of the volatile organic compounds were diverse among the four tomato varieties (Table 3). “Tasti-Lee”, “Tygress”, “FL 47”, and “Cherokee Purple” contained 27, 29, 33, and 31 volatile compounds, respectively. “FL 47” contained the maximum numbers of volatile compounds among these four varieties.

Based on the concentrations of volatile compounds identified in different tissues and per tissue’s percentage to total weight, the whole-fruit volatile profile of these four varieties were calculated (Table 4). As shown in Table 4, all four varieties showed high levels of cis-3-hexenal, hexanal, trans-2-hexenal and acetone, especially cis-3-hexenal in the whole fruit, which imparted leafy and green notes into the tomato fruit. The total concentrations of volatile compounds in “Tygress” was higher than the other three varieties. It contained the highest levels of aldehydes, ketones, and oxygen-containing heterocyclic compounds in the forms of 2-methyl propanal, 2-methyl-2-butenal, cis-3-hexenal, trans-2-hexenal, 1-penten-3-one, and geranyl acetone.

On the other hand, “FL 47” showed the highest levels of alcohols, esters, and nitrogen compounds in the forms of 2-methyl propanol, 3-methyl butanol, 2-methyl butanol, 4-methyl pentanol, 3-methyl pentanol, butyl acetate, and 1-nitro-pentane. In addition, 1-nitro-pentane was unique to “FL 47”, which possessed a pleasant fruity odor (Table 2). Besides, nonanal was detected only in “FL 47” in a low concentration, which imparted fatty, citrusy, and green notes into the tomato fruit.

Only 27 volatile compounds were identified in the “Tasti-Lee” type, and the total concentration of the volatile compounds was lower than other three varieties due to its low concentrations of aldehydes, alcohol, and sulfur- and nitrogen-containing heterocyclic compounds. No butanal and octanal was detected in “Tasti-Lee”, which imparted green and fatty notes into the tomato fruit (Table 2 and Table 4).

Methyl salicylate and 1-nitro-2-phenylethane compounds were found only in the “FL 47” and “Cherokee Purple” fruits, except “Tasti-Lee” and “Tygress”. The concentrations of the volatile compounds (aldehydes, alcohols, and total volatile compounds) in the “Cherokee Purple” fruit were higher than Tasti-Lee. The concentrations of hydrocarbons were lower than the other three varieties. However, these four varieties mainly differed in their content of aldehydes, alcohols, and esters.

Various factors influence the volatile characteristics of different cultivars. Rambla et al. [24] reported that the volatile components were different in 152 “Heirloom” varieties, with different genetic compositions. The difference in volatile characteristics between these four varieties might be due to different genotypes. Conversely, Nesbitt and Tanksley [25] showed a huge heterogeneity in the color, size, shape, and chemical composition of the fruit among the old, open-pollinated “Heirloom” tomato varieties. Paradoxically, DNA sequencing results found negligible polymorphism within the species. The result indicated that the volatile content was affected by many other factors. Besides genetic makeup, the differences in the volatile profile and fruit quality among the cultivars were dependent upon the environment, management, fertilizers, and harvest maturity [26,27,28,29].

### 2.4. Differences in the Volatile Profiles among the Cultivars

According to their odor descriptions as shown in Table 2, these volatile compounds can be divided into six aromatic series, including those which are herbaceous, floral, fruity, fatty, spicy, and like cocoa. The proportion of aroma and its aromatic series per tissue were established based on odor activity values (OAVs) (Figure 2).

#### 2.4.1. The Volatile Profile of Pericarp (PE)

Among the four varieties (“FL 47”, “Tygress”, “Tasti-Lee”, and “Cherokee Purple”), the proportion of total volatile concentrations in PE were 28.34%, 30.81%, 28.42%, and 29.99%, respectively, which showed the highest concentration of the volatile compounds compared to other tissues in all four varieties (Table 5). The concentrations of aldehydes in PE of “Tasti-Lee”, “Tygress”, “FL 47”, and “Cherokee Purple” tomato fruits were 30.56%, 32.40%, 30.59%, and 32.94%, respectively, higher than those in other inner tissues. The aldehydes/alcohols ratio in PE was higher than the inner tissues in SC and LS (Table 4). Aldehydes and their corresponding alcohols were important tomato flavor volatiles. Alcohols could be oxidized to aldehydes, and accomplished the alcohol-to-aldehyde conversion. The high ratio of aldehydes/alcohols in PE may be due to its higher oxygen concentrations in external tissues.

In addition, PE showed higher levels of aromatic proportion in the “Tygress”, “Tasti-Lee”, and “Cherokee Purple” types, which suggested that PE provided a more powerful aroma than the other tissues (Figure 2). The herbaceous series showed a higher intensity in PE, which is due to the high OAVs of aldehydes, especially cis-3-hexenal, hexanal, and trans-2-hexenal. These three compounds were C6-aldehydes which imparted “green”, “leafy”, “grassy”, “tallow”, and “fatty” notes in tomato fruit. Based on their biosynthetic pathways, they were generated from C18 fatty acid, which were acted upon by TomLox C and 13-hydroperoxide lyase (13-HPL). Firstly, 13-hydroperoxides (13-HPOs) were produced by the act of TomLox C, and then they were cleaved by 13-hydroperoxide lyase (13-HPL), which is a key enzyme for C6-aldehydes synthesis to release C6-aldehydes, both hexanal and cis-3-hexenal. The latter could further be converted into trans-2-hexenal [30].

#### 2.4.2. The Volatile Profile of Septa and Columella (SC) and Locular Gel and Seeds (LS)

Then the proportion of total volatile concentrations flowed by SC and LS. According to Table 3, the levels of 2-methyl propanol, 3-methyl butanol, and 2-methyl butanol were high in LS of “Tasti-Lee,” “Tygress,” “FL 47” and “Cherokee Purple” fruits (Table 3), which could impart “alcoholic,” “grassy,” “sweet,” “whiskey,” “malt,” “burnt,” “malt,” “wine,” and “onion” notes in the tomato fruits (Table 2). It was in consistent with results of Wang et al. [16]. Besides, the distribution of other volatile compounds within the four tissues varied among the cultivars. For “FL 47” and “Cherokee Purple” fruit, the SC tissue possessed higher levels of 4-methyl pentanol and 3-methyl pentanol. Conversely, 1-penten-3-one, 6-methyl-5-hepten-2-one, 2-ethyl furan and butyl acetate were present higher concentrations in SC tissue of “Tasti-Lee”.

The aromatic proportion and aroma series of SC and LS behaved significantly differently in different cultivars. The cocoa series showed a higher intensity in LS from “Tasti-Lee” and “Tygress” and in SC from “FL47” (Figure 2). As shown in Table 2, the cocoa flavor was mainly imparted by 2-methyl furan and 2-methyl butanal. The SC of “Tasti-Lee” showed high levels of floral and fruity series compared with other tissues, while these series were present in high levels in LS of “Tygress” and “Cherokee Purple” (Figure 2). The results indicated that floral and fruity series showed a higher intensity in SC and LS. Floral aroma was produced by 6-methyl-5-hepten-2-one, geranyl acetone, and 1-penten-3-one. In tomato fruit, 1-penten-3-one was a fatty-acid-derived volatile. 6-methyl-5-hepten-2-one and geranyl acetone were apocarotenoid volatiles which could be directly synthesized from their carotenoid precursors by the action of carotenoid cleavage dioxygenases (CCD) [1,31]. 6-methyl-5-hepten-2-one directly comes from lycopene, and ζ-carotenoid is the direct precursor for geranylactone [4,32].

In addition to floral notes, they also made considerable contributions to fruity flavor notes (Table 2). 2-methyl-2-butenal, butyl acetate, 2-methylbutyl acetate, and 1-nitro-pentane were also main contributors to the fruity series. Furthermore, previous studies found that the fruity aroma could also be strengthened in some octanal [33], nonanal [34], and limonene [35,36] grape samples due to their lemony flavor, which could be found in our study as well.

#### 2.4.3. The Volatile Profile of Stem End (SE)

The total number of volatile compounds in SE was lesser than the other tissues, and the aromatic proportion of SE from “Tasti-Lee”, “Tygress”, “FL47”, and “Cherokee Purple” were lower than other tissues by 22.23%, 15.72%, 20.37%, and 23.27%, respectively, which suggested that SE provided a less powerful aroma than the other parts (Figure 2). Besides, higher levels of fatty series were shown in SE from “Tygress”, “Tasti-Lee”, and “FL47”, with the existence of that which were hexanal, heptanal, and octanal. Wu et al. [36] found that the fatty series in grapes were contributed mainly by the octanal one, which was consistent with our results.

However, the concentrations of cis-3-hexenal and benzaldehyde were lesser in the SE than the other tissues. Additionally, geranyl acetone, 1-nitro-pentane, and 1-nitro-2-phenylethane, which conferred a “sweet”, “fruity”, “flowery”, and “spicy” odor to tomato fruits, were not identified in SE. 1-nitro-2-phenylethane was a phenylalanine-derived volatile, and an important contributor to the tomato aroma as well. In tomatoes, the first and rate-limiting step was performed by aromatic amino acid decarboxylases (AADCs), encoded by *LeAADC1A, LeAADC1B*, and *LeAADC2* [37]. These enzymes converted phenylalanine to phenethylamine, then phenethylamine was converted to phenylacetaldehyde by an as-yet-unidentified amine oxidase or to 1-nitro-2-phenethane by an uncharacterized series of reactions [1]. The first and rate-limiting step was regulated at the transcriptional level [38], which suggests that the reason why 1-nitro-2-phenylethane was not identified in SE might be due to lower transcriptional levels of *LeAADC1A, LeAADC1B*, and *LeAADC2*.

#### 2.4.4. Principal Component Analysis (PCA) of Volatile Concentration of Different Tissues

PCA was performed to analyze the variation of the 13 important volatile compounds in the four cultivars of tomato, and the results are shown in Figure 3. As shown in this Figure, the first principal component (PC1) and the second principal component (PC2) accounted for 58.3% (Figure 3a), 60.1% (Figure 3b), 62.8% (Figure 3c), and 63.4% (Figure 3d) of the total variance in “Tasti-Lee”, “Tygress”, “FL47”, and “Cherokee Purple”, respectively. PE and SC were separated from other tissues in “Tygress”, “FL47”, and “Cherokee Purple”, which indicated that their flavors were different from the others (Figure 3b–d). SC and LS were separated from PE and SE, but were close to each other in “Tygress” and “FL47”, which indicated that they possessed similar overall flavor (Figure 3b,c). Only LS could be separated from others in “Tasti-Lee”, while on the contrary, all tissues could be separated from each other in “Cherokee Purple”. Six volatile compounds, including hexanal, 3-methyl butanol, 2-methyl butanol, 6-methyl-5-hepten-2-one, 3-methyl butanal, and 2-methyl butanal were present in sufficient quantities so as to influence the tomato flavor.

These volatile compounds were biosynthesized from various pathways in the tomato. Hexanal was generated from C18 fatty acids [30], which conferred “grassy”, “tallow”, “fatty” notes to the tomato. Also, 3-methyl butanol and 2-methyl butanol were biosynthesized via the removal of amino groups from amino acids by branched chain aminotransferases (BCATs). Subsequently, the aldehydes were produced via decarboxylation, which were then reduced to form alcohols [39]. *LeCCD1A* and *LeCCD1B* (carotenoid cleavage dioxygenases) cleaved the carotenoids to synthesize 6-methyl-5-hepten-2-one in the tomato [31]. Also, 3-methyl butanal and 2-methyl butanal, which imparted “malty”, “cocoa”, and “almond” notes to the tomato, were biosynthesized from the amino acids (Table 2).

## 3. Materials and Methods

### 3.1. Plant Materials

Four different varieties of fresh tomato fruits, including “Tygress”, “Tasti-Lee”, “Cherokee Purple”, and “FL 47”, were harvested at the fully ripe stage from a tomato research block at the USDA Picos Road Farm in Fort Pierce, Florida, USA. For each cultivar, 30 defect-free and uniform fruits were divided into three groups to represent three biological replicates.

### 3.2. Methods

#### 3.2.1. Sample Processing

The fruit were separated into the following four tissues: pericarp (PE), septa and columella (SC), locular gel and seeds (LS), and stem end (SE), by using a sharp stainless-steel knife (Figure 1). The tissues were rapidly immersed in liquid nitrogen, ground to a powder, and the resulting 4.3 g of tissue powder together with 1.7 mL of saturated CaCl_2_ solution were transferred to a 20 mL vial sealed with Teflon-lined septa to be smashed, and finally stored at −80 °C until analysis.

#### 3.2.2. Analysis of Volatile Components using Headspace Gas Chromatography-Mass Spectrometry

The solid-phase micro-extraction-gas chromatography-mass spectrometry (SPME-GC-MS) analysis was conducted following our previous studies [40,41] with some modifications. Volatiles were extracted using an SPME fiber (50/30 μm DVB/Carboxen/PDMS; Supelco, Bellefonte, PA, USA). The SPME fiber was put into the headspace vial, and 1 cm of it was exposed from the headspace for 40 min at 50 °C. After extraction, the fiber was inserted into the injector of a GC-MS (Model 6890; Agilent, Santa Clara, CA, USA) to desorb the adsorbed substances for 5 min at 250 °C. At the same time, the instrument data acquisition was performed.

Gas chromatography was performed using the HP-5 column (50 m × 0.32 mm × 1.05 μm, J&W Scientific, Agilent, Santa Clara, CA, USA) with helium as the carrier gas (37 kPa). The column temperature was set at 40 °C for 2 min, then increased to 250 °C at the rate of 5 °C min^−1^, and finally maintained at 250 °C for the next 2 min. The volatile compounds were matched against the NIST08 library (NIST/EPA/NIH, American), and the retention indexes were compared with the standard volatile compounds. A standard peak area vs. concentration curve was prepared from the serial dilutions of the standard and used for sample quantification.

#### 3.2.3. Statistical Analysis

All quantifications were carried out with five biological replicates, and the data of the study results were expressed as the average of five replicates. Principal component analysis (PCA) was performed using JMP 11.2.0 software (SAS Institute, Cary, NC, USA) on the covariance for analyzing the significant differences and relationships of the volatile organic compounds among the different tissues. The data were analyzed using the Statistical Analysis System Version 9.3 (SAS Institute, Cary, NC, USA). The volatile concentrations between different cultivars and different tissues were analyzed using the analysis of variance (ANOVA). The mean separation was determined by Duncan’s test at a significance level of 5 %, respectively.

## 4. Conclusions

In this study, a total of 36 volatile compounds were detected in four varieties of tomatoes which, chemically, were aldehydes, hydrocarbons, alcohols, ketones, furans, esters, nitrogen compounds, and sulfur and nitrogen-containing heterocyclic compounds. The results of our study showed that the content of the volatile compounds varied among the four cultivars of tomato. “FL 47” contained more volatile compounds than the other three varieties, and “Tygress” fruit possessed the highest levels of aldehydes, ketones, and oxygen-containing heterocyclic compounds. The volatile compositions of pericarp (PE), septa and columella (SC), locular gel and seeds (LS), and stem end (SE) tissues were quite different. The abundance of total volatile compounds was higher in PE, which was associated with higher levels of aldehydes. Meanwhile, it showed higher aromatic proportion and herbaceous series intensity than other tissues. SC and LS showed a higher intensity of floral and fruity series. The concentration of alcohols in LS was higher than that in the other tissues, in association with the higher abundances of 2-methyl propanol, 3-methyl butanol, and 2-methyl butanol. The total volatile concentration and aromatic proportion in SE were lesser than the other tissues. Additionally, higher levels of fatty series were shown in SE, and the floral and fruity characteristic compounds of geranyl acetone, 1-nitro-pentane, and 1-nitro-2-phenylethane were not identified in SE.

## Figures and Tables

**Figure 1 molecules-24-02594-f001:**
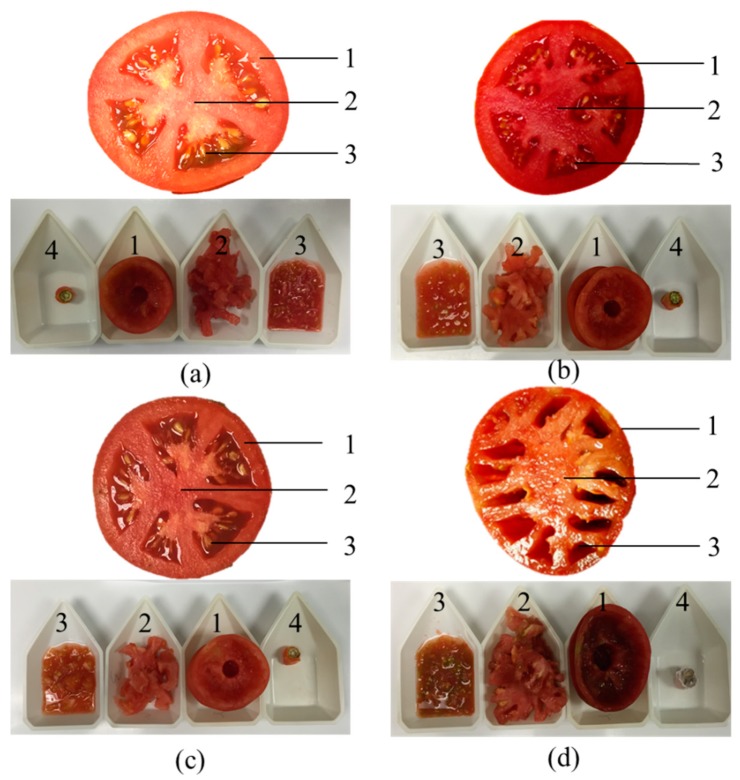
Schematic cross-section of (**a**) Tasti-Lee, (**b**) Tygreen, (**c**) FL47, and (**d**) Cherokee Purplea tomato fruit. (1) Pericarp (PE); (2) septa and columella (SC); (3) locular gel and seeds (LS); and (4) stem end (SE).

**Figure 2 molecules-24-02594-f002:**
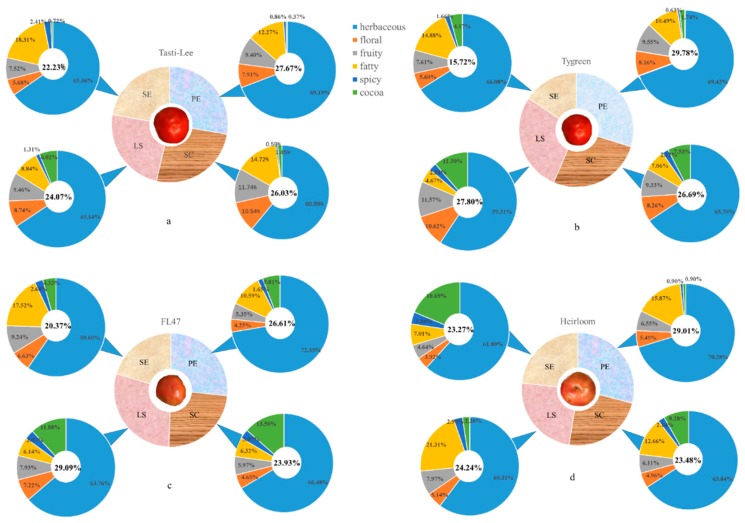
Aromatic proportion and aromatic series per tissue in tomatoes. (**a**) “Tasti-Lee”, (**b**) “Tygress”, (**c**) “FL47”, and (**d**) “Cherokee Purple”.

**Figure 3 molecules-24-02594-f003:**
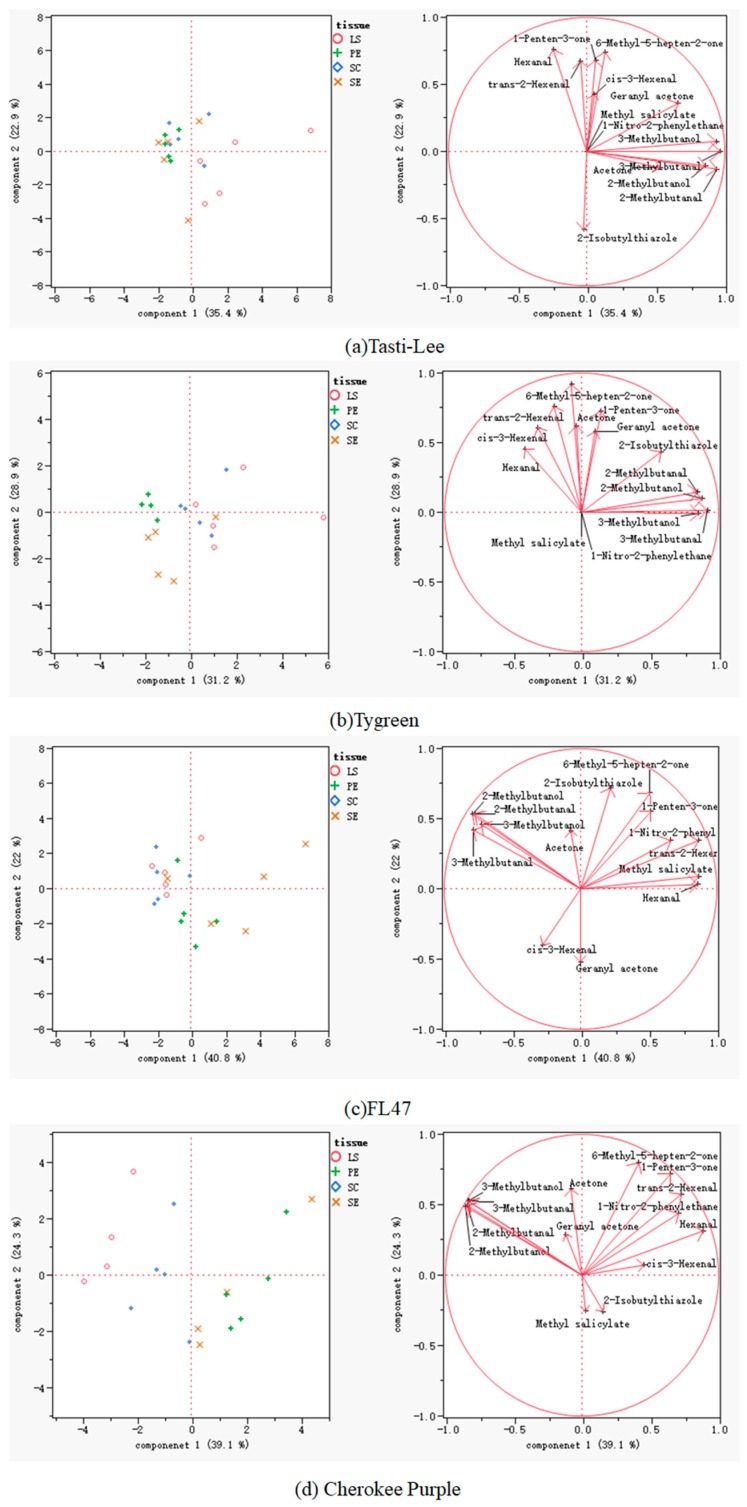
Principal component analysis (PCA) result of different tissues.

**Table 1 molecules-24-02594-t001:** Percentage to total weight per tissue in four tomato varieties.

Variety	Weight (g)	Percentage to Total Weight (%)			
		Pericarp (PE)	Septa and Columella (SC)	Locular Gel and Seeds (LS)	Stem End (SE)
FL47	206.10	52.83 ± 5.37 ^a^	29.70 ± 2.55 ^b^	16.26 ± 3.86 ^c^	1.21 ± 0.27 ^d^
Tygress	181.85	54.36 ± 4.93 ^a^	27.18 ± 3.37 ^b^	17.02 ± 2.03 ^c^	1.44 ± 0.26 ^d^
Tasti-Lee	159.35	53.99 ± 2.51 ^a^	27.53 ± 1.67 ^b^	16.76 ± 2.07 ^c^	1.73 ± 0.19 ^d^
Cherokee Purple	265.58	47.34 ± 3.33 ^a^	40.24 ± 3.76 ^b^	10.09 ± 1.60 ^c^	2.33 ± 0.60 ^d^

Values are expressed as means ± standard deviation. Different superscripts (a–d) in the same row indicate significant difference (*p* < 0.05).

**Table 2 molecules-24-02594-t002:** Identification of volatile compounds identified in tomato fruits, along with their odor descriptions, odor thresholds in water, and RI values.

	Compounds	Odor Description	Odor Threshold in Water (mg L^−1^)	RI
**Aldehydes**				
1	2-Methylpropanal	pungent, malt, green	0.0009–0.001	567
2	Butanal	Pungent, green	0.009	590
3	3-Methylbutanal	Malt	0.00015–0.0002	638
4	2-Methylbutanal	Cocoa, almond, malt	0.003	646
5	2-Methyl-2-butenal	Green, fruit	0.5	719
6	cis-3-Hexenal	Leafy, green	0.00025	771
7	Hexanal	Grass, tallow, fat	0.0045–0.005	774
8	trans-2-Hexenal	Green, leafy	0.017	828
9	Heptanal	Fat, citrus, rancid	0.003	875
10	trans, trans-2, 4-Hexadienal	Green	0.06	887
11	Benzaldehyde	Almond, burnt sugar	0.35	945
12	Octanal	Fat, soap, lemon, green	0.0007	969
13	Nonanal	Fat, citrus, green	0.001	1059
**Hydrocarbons**				
1	Cymene	Solvent, gasoline, citrus	0.15	994
2	Limonene	Lemon, orange	0.01	998
3	Terpinolene	Smokey, woody	0.2	1048
4	Undecane	Alkane	10	1051
5	Dodecane	Alkane	10	1137
6	Tridecane	Alkane		1222
**Alcohols**				
1	2-Methylpropanol	Alcoholic, grassy, sweet	12.5	612
2	3-Methylbutanol	Whiskey, malt, burnt	0.25–0.3	707
3	2-Methylbutanol	Malt, wine, onion	0.25–0.3	711
4	4-Methylpentanol	Pungent	0.82–4.1	809
5	3-Methylpentanol	Pungent	0.83–4.1	817
**Ketones**				
1	Acetone	Pungent, irritating, floral	40	533
2	1-Penten-3-one	Fruity, floral, green	0.0015	665
3	6-Methyl-5-hepten-2-one	Fruity, floral	0.05	950
4	Geranyl acetone	Sweet, floral, estery	0.06	1367
**Oxygen-containing heterocyclic compounds**				
1	2-Methylfuran	Chocolate	0.2	602
2	2-Ethyl furan	Rum, coffee and chocolate	-	676
**Esters**				
1	Butyl acetate	Pear	0.066	744
2	2-Methylbutyl acetate	Fruit	0.005–0.011	847
3	Methyl salicylate	Peppermint	0.04	1156
**Nitrogen compounds**				
1	1-Nitro-pentane	Pleasant, fruity	22	916
**Sulfur- and nitrogen-containing heterocyclic compounds**				
1	2-Isobutylthiazole	Tomato leafy, green	0.0035	1002
2	1-Nitro-2-phenylethane	Flower, spice	0.002	1250

-, no data was reported.

**Table 3 molecules-24-02594-t003:** Concentrations of volatile compounds identified in different tissues of tomato fruits.

Compounds	Concentration (mg L^−1^)															
	Tasti-Lee				Tygress				FL47				Cherokee Purple			
	PE	SC	LS	SE	PE	SC	LS	SE	PE	SC	LS	SE	PE	SC	LS	SE
**Aldehydes**																
2-Methylpropanal	- ^z^	-	0.015 ^a, y^	0.037 ^a^	-	0.054 ^b^	0.082 ^c^	-	-	0.047 ^b^	0.067 ^b^	0.011 ^a^	-	0.022 ^b^	-	0.076 ^c^
Butanal	-	-	-	-	-	0.029 ^a^	0.044 ^a^	0.010 ^a^	0.016 ^a^	0.025 ^a^	0.059 ^a^	0.043 ^a^	0.010 ^a^	0.028 ^b^	0.012 ^a^	0.146 ^c^
3-Methylbutanal	0.082 ^a^	0.320 ^a^	1.303 ^b^	0.156 ^a^	0.269 ^a^	0.745 ^a,b^	1.709 ^c^	0.451 ^a^	2.265 ^a,b^	3.643 ^b^	3.530 ^b^	0.959 ^a^	0.279 ^a^	1.200 ^b^	0.371 ^a^	1.930 ^c^
2-Methylbutanal	0.054 ^a^	0.217 ^a^	0.748 ^b^	0.084 ^a^	0.362 ^a^	1.394 ^b^	2.158 ^c^	0.463 ^a^	0.984 ^a^	2.082 ^b^	2.273 ^b^	0.655 ^a^	0.146 ^a^	1.131 ^b^	0.294 ^a^	2.549 ^c^
2-Methyl-2-butenal	0.006 ^a^	0.009 ^a^	0.006 ^a^	0.008 ^a^	0.040 ^a^	0.415 ^b^	0.470 ^b^	0.065 ^a^	0.079 ^a^	0.009 ^a^	0.004 ^a^	0.003 ^a^	0.008 ^a^	0.019 ^a^	0.005 ^a^	0.023 ^a^
cis-3-Hexenal	14.538 ^b^	9.295 ^a,b^	7.548 ^a^	6.913 ^a^	17.458 ^b^	14.024 ^b^	9.104 ^a^	6.186 ^a^	13.339 ^c^	6.608 ^b^	8.014 ^b^	4.268 ^a^	12.955 ^b^	8.127 ^a^	5.067 ^a^	4.220 ^a^
Hexanal	2.763 ^a,b^	3.448 ^b^	1.726 ^a^	3.517 ^b^	3.238 ^c^	1.835 ^a,b^	1.295 ^a^	2.383 ^b,c^	2.872 ^a,b^	1.432 ^a^	1.844 ^a^	3.935 ^b^	4.235 ^b,c^	2.566 ^a,b^	4.876 ^b^	1.342 ^a^
trans-2-Hexenal	2.393 ^a^	3.142 ^a^	2.754 ^a^	3.830 ^a^	5.072 ^a^	3.962 ^a^	4.886 ^a^	3.370 ^a^	3.011 ^a,b^	2.282 ^a^	4.066 ^a,b^	5.252 ^b^	4.698 ^a^	2.724 ^a^	5.485 ^a^	3.290 ^a^
Heptanal	0.031 ^b,c^	0.017 ^a,b^	0.012 ^a^	0.045 ^c^	0.027 ^a,b^	0.030 ^b^	0.016 ^a^	0.031 ^b^	0.029 ^a^	0.021 ^a^	0.016 ^a^	0.060 ^b^	0.028 ^a^	0.022 ^a^	0.073 ^b^	0.012 ^a^
trans, trans-2,4-Hexadienal	0.103 ^a^	0.083 ^a^	0.073 ^a^	0.068 ^a^	-	-	-	-	-	-	-	0.065 ^a^	0.114 ^a^	0.051 ^a^	0.089 ^a^	-
Benzaldehyde	0.015 ^a^	0.015 ^a^	0.008 ^a^	0.007 ^a^	0.020 ^a^	0.024 ^a^	0.011 ^a^	0.009 ^a^	0.020 ^a^	0.019 ^a^	0.016 ^a^	0.009 ^a^	0.018 ^a^	0.016 ^a^	0.011 ^a^	0.006 ^a^
Octanal	-	-	-	-	0.001 ^a^	0.0002 ^a^	0.0004 ^a^	0.003 ^a^	0.002 ^a^	0.002 ^a^	0.0007 ^a^	0.005 ^b^	0.004 ^a^	0.003 ^a^	0.008 ^b^	0.002 ^a^
Nonanal	-	-	-	-	-	-	-	-	-	-	-	0.0024 ^a^	-	-	-	-
**Hydrocarbons**																
Cymene	-	-	-	-	0.0002 ^a^	-	0.00008 ^a^	0.00009 ^a^	-	-	-	-	-	0.00004 ^a^	-	-
Limonene	0.034 ^a^	0.038 ^a^	0.014 ^a^	0.019 ^a^	0.042 ^a^	0.002 ^a^	0.017 ^a^	0.015 ^a^	0.010 ^a^	0.013^a^	0.014 ^a,b^	0.009 ^a^	0.004 ^a^	0.005 ^a^	0.013 ^a^	0.005 ^a^
Terpinolene	-	-	-	-	0.0004 a	-	-	-	-	-	-	-	-	-	-	-
Undecane	0.024 ^a^	0.010 ^a^	0.002 ^a^	-	-	-	-	-	-	-	-	-	-	-	-	-
Dodecane	0.054 ^b^	0.009 ^a^	-	-	-	-	-	-	0.0024 ^a^	0 ^a^	0.005 ^a^	-	-	0.012 ^a^	0.010 ^a^	0.015 ^a^
Tridecane	0.089 ^a^	0.043 ^a^	0.019 ^a^	0.016 ^a^	-	0.015 ^a^	-	0.014 ^a^	-	0.019^a^	-	-	-	-	-	-
**Alcohols**																
2-Methylpropanol	-	-	0.013 ^a^	-	0.007 ^a^	0.097 ^b^	0.110 ^b^	0.018 ^a^	0.057 ^a^	0.096 ^a^	0.096 ^a^	0.036 ^a^	-	0.043 ^b^	0.112 ^c^	0.004 ^a^
3-Methylbutanol	0.078 ^a^	0.238 ^a^	1.141 ^b^	0.258 ^a^	0.198 ^a^	0.498 ^a^	1.446 ^a^	0.507 ^a^	2.666 ^a,b^	3.79 ^b,c^	4.507 ^c^	1.529 ^a^	0.411 ^a^	1.312 ^b^	2.248 ^c^	0.593 ^a^
2-Methylbutanol	0.054 ^a^	0.197 ^a,b^	0.370 ^b^	0.195 ^a,b^	0.302 ^a^	0.820 ^b^	0.929 ^b^	0.474 ^a^	0.753 ^a,b^	1.101 ^b^	1.184 ^b^	0.586 ^a^	0.294 ^a^	0.876 ^b^	1.290 ^c^	0.421 ^a^
4-Methylpentanol	-	-	0.023 ^a^	0.108 ^a^	-	0.051 ^a^	0.036 ^a^	0.062 ^a^	0.097 ^a,b^	0.134 ^b^	0.051 ^a^	0.114 ^a,b^	0.042 ^a^	0.090 ^b,a^	0.065 ^a^	0.034 ^a^
4-Methylpentanol	0.007 ^a^	0.013 ^a^	0.032 ^a^	0.009 ^a^	0.017 ^a^	0.026 ^a^	0.055 ^a^	0.024 ^a^	0.112 ^a^	0.152 ^a^	0.093 ^a^	0.096 ^a^	0.021 ^a^	0.041 ^b^	0.025 ^a^	0.024 ^a^
**Ketones**																
Acetone	0.813 ^a^	1.179 ^a,b^	2.368 ^b^	1.169 ^a,b^	2.067 ^a^	1.822 ^a^	1.474 ^a^	2.084 ^a^	1.708 ^a^	2.785 ^a^	1.827 ^a^	1.678 ^a^	1.345 ^a^	1.206 ^a^	1.210 ^a^	1.791 ^a^
1-Penten-3-one	0.090 ^a,b^	0.118 ^b^	0.082 ^a,b^	0.055 ^a^	0.120 ^b^	0.108 ^a,b^	0.149 ^b^	0.047 ^a^	0.055 ^a^	0.056 ^a^	0.100 ^a^	0.080 ^a^	0.065 ^a^	0.048 ^a^	0.055 ^a^	0.038 ^a^
6-Methyl-5-hepten-2-one	0.035 ^a^	0.042 ^a^	0.026 ^a^	0.030 ^a^	0.053 ^a^	0.037 ^a^	0.033 ^a^	0.025 ^a^	0.025 ^a^	0.021 ^a^	0.026 ^a^	0.033 ^a^	0.043 ^a^	0.028 ^a^	0.034 ^a^	0.036 ^a^
Geranyl acetone	-	0.138 ^a^	0.139 ^a^	-	0.390 ^b^	0.224 ^a,b^	0.103 ^a,b^	-	0.230 ^a^	-	-	-	-	0.218 ^a^	-	-
**Oxygen-containing heterocyclic compounds**																
2-Methylfuran	0.019 ^a^	0.031 ^a,b^	0.044 ^b^	0.020 ^a^	0.046 ^a,b^	0.067 ^b^	0.061 ^b^	0.033 ^a^	0.057 ^b^	0.076 ^b^	0.058 ^b^	0.017 ^a^	0.024 ^b^	0.045 ^c^	0.010 ^a^	0.029 ^b^
2-Ethyl furan	0.024 ^a^	0.028 ^a^	0.017 ^a^	0.020 ^a^	0.035 ^a^	0.025 ^a^	0.022 ^a^	0.016 ^a^	0.017 ^a^	0.014 ^a^	0.017 ^a^	0.022 ^a^	0.029 ^a^	0.019 ^a^	0.023 ^a^	0.024^a^
**Esters**																
Butyl acetate	-	0.006 ^b^	0.006 ^b^	0.005 ^b^	0.008 ^a^	0.011 ^a^	0.010 ^a^	0.009 ^a^	0.008 ^a^	0.014 ^b,c^	0.016 ^c^	0.010 ^a,b^	0.003 ^a,b^	0.008 ^b^	0.005 ^a,b^	0.002 ^a^
2-Methylbutyl acetate	-	-	-	-	0.004 ^b^	0.004 ^b^	0.003 ^a,b^	0.001 ^a^	0.002 ^a,b^	0.003 ^b^	0.001 ^a^	0.0003 ^a^	0.0004 ^ab^	0.001 ^a,b^	0.002 ^b^	-
Methyl salicylate	-	-	-	-	-	-	-	-	-	-	-	0.018 ^b^	-	-	0.008 ^a^	-
**Nitrogen compounds**																
1-Nitro-pentane	-	-	-	-	-	-	-	-	0.006 ^a,b^	0.009 ^b^	0.005 ^a,b^	-	-	-	-	-
**Sulfur-and Nitrogen-containing heterocyclic compounds**																
2-Isobutylthiazole	-	-	-	0.0003 ^a^	0.007 ^a^	0.013 ^a^	0.012 ^a^	-	0.005 ^a^	0.013 ^a^	0.006 ^a^	0.009 ^a^	0.0006 ^a^	-	-	-
1-Nitro-2-phenylethane	-	-	-	-	-	-	-	-	-	-	0.020 ^a^	-	0.018 ^a^	-	0.017 ^a^	-
**Sum**																
Aldehydes	19.984 ^a^	16.546 ^a^	14.191 ^a^	14.664 ^a^	26.486 ^c^	22.512 ^b,c^	19.775 ^b^	12.971 ^a^	22.616 ^c^	16.170 ^a,b^	19.890 ^b,c^	15.269 ^a^	22.495 ^b^	15.909 ^a,b^	16.290 ^a,b^	13.595 ^a^
Hydrocarbons	0.202 ^b^	0.100 ^a^	0.036 ^a^	0.035 ^a^	0.043 ^a^	0.017 ^a^	0.017 ^a^	0.029 ^a^	0.013 ^a^	0.032 ^a^	0.019 ^a^	0.009 ^a^	0.004 ^a^	0.017 ^a^	0.023 ^a^	0.019 ^a^
Alcohols	0.139 ^a^	0.448 ^a,b^	1.578 ^b^	0.569 ^a,b^	0.523 ^a^	1.493 ^a,b^	2.576 ^b^	1.085 ^a,b^	3.684 ^a,b^	5.272 ^b,c^	5.931 ^c^	2.362 ^a^	0.769 ^a^	2.361 ^b^	3.739 ^c^	1.075 ^a^
Ketones	0.945 ^a^	1.469 ^a,b^	2.615 ^b^	1.254 ^a^	2.630 ^a^	2.192 ^a^	1.759 ^a^	2.156 ^a^	2.018 ^a^	2.861 ^a^	1.952 ^a^	1.791 ^a^	1.453 ^a^	1.501 ^a^	1.299 ^a^	1.865 ^a^
Oxygen-containing heterocyclic compounds	0.047 ^a^	0.054 ^a^	0.062 ^a^	0.039 ^a^	0.082 ^a,b^	0.091 ^b^	0.082 ^a,b^	0.050 ^a^	0.073 ^b^	0.090 ^b^	0.075 ^b^	0.040 ^a^	0.052 ^a,b^	0.063 ^b^	0.033 ^a^	0.053 ^a,b^
Esters	-	0.006 ^b^	0.006 ^b^	0.005 ^b^	0.012 ^a^	0.016 ^a^	0.013 ^a^	0.010 ^a^	0.010 ^b^	0.017 ^a,b^	0.017 ^a,b^	0.028 ^b^	0.004 ^a,b^	0.009 ^a,b^	0.014 ^b^	0.002 ^a^
Nitrogen compounds	-	-	-	-	-	-	-	-	0.006 ^a,b^	0.009 ^b^	0.005 ^a,b^	-	-	-	-	-
Sulfur- and nitrogen-containing heterocyclic compounds	-	-	-	0.0003 ^a^	0.007 ^a^	0.013 ^a^	0.012 ^a^	-	0.005 ^a^	0.013 ^a^	0.006 ^a^	0.030 ^a^	0.019 ^a^	-	0.017 ^a^	-
Total compounds	21.317 ^a^	18.624 ^a^	18.486 ^a^	16.567 ^a^	29.781 ^b^	26.333 ^b^	24.234 ^b^	16.300 ^a^	28.425 ^b^	24.464 ^a,b^	27.895 ^b^	19.528 ^a^	24.795 ^a^	19.860 ^a^	21.414 ^a^	16.608 ^a^
Aldehydes/Alcohols ratio	143.925 ^b^	36.917 ^a^	8.992 ^a^	25.757 ^a^	50.629 ^b^	15.081 ^a^	7.676 ^a^	11.961 ^a^	6.138 ^a^	3.067 ^a^	3.354 ^a^	6.465 ^a^	29.267 ^b^	6.738 ^a^	4.357 ^a^	12.647 ^a^

^z^ -, the volatile compound was not found. ^y^ Different superscripts (a–c) in the same row indicate significant difference (*p* < 0.05).

**Table 4 molecules-24-02594-t004:** Whole-fruit volatile profile of different tomato varieties.

No.	Compounds	Concentration (mg L^−1^)			
		Tasti-Lee	Tygreen	FL47	Cherokee Purple
**Aldehydes**					
1	2-Methyl propanal	0.003 ^a^	0.029 ^b^	0.025 ^b^	0.011 ^a^
2	Butanal	-	0.016 ^a,b^	0.026 ^b^	0.021 ^a,b^
3	3-Methyl butanal	0.353 ^a^	0.646 ^a^	2.864 ^b^	0.697 ^a^
4	2-Methyl butanal	0.215 ^a^	0.950 ^c^	1.516 ^d^	0.614 ^b^
5	2-Methyl-2-butenal	0.007 ^a^	0.215 ^b^	0.045 ^a^	0.013 ^a^
6	cis-3-Hexenal	11.793 ^a,b^	14.940 ^b^	10.364 ^a^	10.013 ^a^
7	Hexanal	2.791 ^a,b^	2.514 ^a^	2.290 ^a^	3.561 ^b^
8	trans-2-Hexenal	2.685 ^a^	4.714 ^b^	2.993 ^a^	3.950 ^a,b^
9	Heptanal	0.024 ^a^	0.026 ^a^	0.025 ^a^	0.030 ^a^
10	trans, trans-2, 4-Hexadienal	0.092 ^a^	-	0.001 ^a^	0.084 ^a^
11	Benzaldehyde	0.014 ^a^	0.019 ^a^	0.019 ^a^	0.016 ^a^
12	Octanal	-	0.001 ^a,b^	0.002 ^b^	0.004 ^c^
13	Nonanal	-	-	0.000029 ^a^	-
**Hydrocarbons**					
1	Cymene	-	0.0001 ^a^	-	-
2	Limonene	0.032 ^a^	0.026 ^a^	0.012 ^a^	0.005 ^a^
3	Terpinolene	-	0.0002 ^a^	-	-
4	Undecane	0.016 ^a,b^	-	-	-
5	Dodecane	0.032 ^b^	-	0.002 ^a^	0.006 ^a^
6	Tridecane	0.064 ^b^	0.004 ^a^	0.006 ^a^	-
**Alcohols**					
1	2-Methyl propanol	0.002 ^a^	0.049 ^c^	0.075 ^d^	0.029 ^b^
2	3-Methyl butanol	0.303 ^a^	0.496 ^a^	3.285 ^b^	0.963 ^a^
3	2-Methyl butanol	0.149 ^a^	0.552 ^b^	0.924 ^c^	0.632 ^b^
4	4-Methyl pentanol	0.006 ^a^	0.021 ^a^	0.101 ^b^	0.063 ^b^
5	3-Methyl pentanol	0.013 ^a^	0.026 ^a^	0.121 ^b^	0.029 ^a^
**Ketones**					
1	Acetone	1.181 ^a^	1.90 ^a^	2.047 ^a^	1.286 ^a^
2	1-Penten-3-one	0.096 ^a,b^	0.121 ^b^	0.063 ^a^	0.057 ^a^
3	6-Methyl-5-hepten-2-one	0.036 ^a^	0.045 ^a^	0.024 ^a^	0.036 ^a^
4	Geranyl acetone	0.061 ^a^	0.290 ^a^	0.122 ^a^	0.088 ^a^
**Oxygen-containing heterocyclic compounds**					
1	2-Methyl furan	0.026 ^a^	0.054 ^b^	0.062 ^b^	0.031 ^a^
2	2-Ethyl furan	0.024 ^a^	0.030 ^a^	0.016 ^a^	0.024 ^a^
**Esters**					
1	Butyl acetate	0.003 ^a^	0.009 ^b,c^	0.011 ^c^	0.005 ^a,b^
2	2-Methylbutyl acetate	-	0.004 ^c^	0.002 ^b^	0.0007 ^a^
3	Methyl salicylate	-	-	0.0002 ^a^	0.0008 ^a^
**Nitrogen compounds**					
1	1-Nitro-pentane	-	-	0.007^b^	-
**Sulfur- and nitrogen-containing heterocyclic compounds**					
1	2-Isobutylthiazole	0.000006 ^a^	0.009 ^a^	0.008 ^a^	0.0003 ^a^
2	1-Nitro-2-phenylethane	-	-	0.003 ^a^	0.010 ^a^
**Sum**					
	Aldehydes	17.977 ^a^	24.069 ^b^	20.169 ^a,b^	19.012 ^a^
	Hydrocarbons	0.143 ^b^	0.031 ^a^	0.019 ^a^	0.011 ^a^
	Alcohols	0.473 ^a^	1.144 ^a,b^	4.505 ^c^	1.716 ^b^
	Ketones	1.373 ^a^	2.356 ^a^	2.255 ^a^	1.466 ^a^
	Oxygen-containing heterocyclic compounds	0.050 ^a^	0.084 ^a^	0.078 ^a^	0.055 ^a^
	Esters	0.003 ^a^	0.013 ^b^	0.013 ^b^	0.007 ^a^
	Nitrogen compounds	-	-	0.007 ^b^	-
	Sulfur- and nitrogen-containing heterocyclic compounds	0.000006 ^a^	0.009 ^a^	0.011 ^a^	0.011 ^a^
	Total compounds	20.018 ^a^	27.706 ^b^	27.058 ^b^	22.277 ^a,b^

Different superscripts (a–d) in the same raw indicate significant difference (*p* < 0.05); -, the volatile compound was not found.

**Table 5 molecules-24-02594-t005:** Volatile total concentrations per tissue.

Concentration (mg L^−1^)				
	Pericarp (PE)	Septa and Columella (SC)	Locular Gel and Seeds (LS)	Stem End (SE)
FL47	28.425 ± 4.551 ^b^	24.464 ± 3.277 ^a,b^	27.895 ± 1.840 ^b^	19.528 ± 5.069 ^a^
Tygress	29.781 ± 7.373 ^b^	26.333 ± 3.031 ^b^	24.234 ± 4.493 ^b^	16.300 ± 6.411 ^a^
Tasti-Lee	21.317 ± 5.934 ^a^	18.624 ± 2.550 ^a^	18.486 ± 10.662 ^a^	16.567 ± 6.529 ^a^
Cherokee Purple	24.795 ± 4.904 ^a^	19.860 ± 6.902 ^a^	21.414 ± 8.067 ^a^	16.608 ± 5.219 ^a^

Values are expressed as means ± standard deviation. Different superscripts (a and b) in the same raw indicate significant difference (*p* < 0.05).

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
