# Peer review of "Distribution of Volatile Compounds in Different Fruit Structures in Four Tomato Cultivars"

_molecules, 2019, doi:10.3390/molecules24142594_

Round 1

Reviewer 1 Report

In this paper, the distribution of volatiles in fresh tomato fruit is carefully described. Results are interesting. 

In Introduction, please clarify the aspect of the number of volatiles which contribute to the tomato aroma. To the best of my knowledge, this aspect is still discussed. 

Please add more details in materials and methods section:

- what test was used for ANOVA?

-how was the PCA performed? on the variance/covariance or correlation matrix?

- please add a reference for OT values.

Author Response

Point 1: In Introduction, please clarify the aspect of the number of volatiles which contribute to the tomato aroma. To the best of my knowledge, this aspect is still discussed.

Response 1: Thank you for your comments. The number of volatiles which contribute to the tomato aroma is not exactly. “Scientists have identified more than four hundred of the volatile compounds in the tomato fruit [5] but only 30 of them…” in original manuscript was changed to “Scientists have identified more than four hundred of the volatile compounds in the tomato fruit [5] but less than 10% of them…”.

Point 2: Please add more details in materials and methods section:

what test was used for ANOVA?

Response 2: Thank you for your suggestions. I have added the test used for ANOVA. The original manuscript was changed to “The volatile concentrations between different cultivars and different tissues were analyzed using the analysis of variance (ANOVA). The mean separation was determined by Duncan’s test at a significance level of 5 %, respectively.”

Point 3: How was the PCA performed? on the variance/covariance or correlation matrix?

Response 3: I have added more details about PCA. The original manuscript was changed to “Principal component analysis (PCA) was performed using the software of JMP 11.2.0 (SAS Institute, Cary, NC., USA) on the covariance for analysing the significant differences and relationships of the volatile organic compounds among the different tissues”.

Point 4: Please add a reference for OT values.

Response 4: Thanks for your suggestions. I have added the reference for OT values.

Reviewer 2 Report

The manuscript focus on the determination by SPME/GC-MS of volatile compounds from tomato different structures. The study is well structured, the methods are properly described as well as the results and the conclusions.

I would suggest, that the aim of the paper to be a little more strengthened (please mention the originality of the study, what is it added-value compared with other studies, etc.).

Line 96: "39 volatile compounds were present in sufficient quantities in tomato which can influence its flavor...." - in introduction (line 39) the number of volatile compounds that influenced the flavor is mentioned to be 30 - please clarify this aspect

Tables - please put the letters in superscript

Why are the values (in tables 3 and 4) expressed with four decimals? I think 3 decimals are enough and the value of standard deviation mentioned.

Author Response

Point 1: I would suggest, that the aim of the paper to be a little more strengthened (please mention the originality of the study, what is it added-value compared with other studies, etc.).

Response 1: Thanks for your suggestions. I have strengthened the aim of the paper. “The aim of the paper was to analyze the distribution of volatile compounds in different fruit structures in four tomato cultivars and evaluate the aroma contributions and aroma profile of different tissues based on their odor activity values (OAVs), which would provide substantial information regarding the volatile components in different cultivars and different inner tissues.

Point 2: Line 96: "39 volatile compounds were present in sufficient quantities in tomato which can influence its flavor...." - in introduction (line 39) the number of volatile compounds that influenced the flavor is mentioned to be 30 - please clarify this aspect.

Response 2: Thanks for your comments. The number of volatiles that influenced the flavor maybe had little difference in different references, and there is not an exact number. “39 volatile compounds were present in sufficient quantities in tomato which can influence its flavor [4, 21-23], and 13 of them had been identified in our study” and “only 30 of them…” in original manuscript were changed to “13 volatile compounds, which were suggested to be important tomato aroma contributors were identified in our study” and “less than 10% of them” respectively.

Point 3: Tables - please put the letters in superscript

Response 3: Thank you for your suggestions. I have put the letters in superscript.

Point 4: Why are the values (in tables 3 and 4) expressed with four decimals? I think 3 decimals are enough and the value of standard deviation mentioned.

Response 4: Thank you for your suggestions. I have changed the values (in tables 3 and 4) expressed with three decimals.

Round 2

Reviewer 1 Report

The quality of this manuscript has been improved. All issues have been addressed and in my opinion the paper is suitable for publication.